# Improvement in Thermal Storage Effectiveness of Paraffin with Addition of Aluminum Oxide Nanoparticles

**DOI:** 10.3390/ma15134427

**Published:** 2022-06-23

**Authors:** Dawit Gudeta Gunjo, Vinod Kumar Yadav, Devendra Kumar Sinha, Mohamed Abdelghany Elkotb, Gulam Mohammed Sayeed Ahmed, Nazia Hossain, Mostafa A. H. Abdelmohimen

**Affiliations:** 1Department of Mechanical Engineering, Adama Science and Technology University, Adama 1888, Ethiopia; dawitphd@gmail.com (D.G.G.); ds3621781@gmail.com (D.K.S.); drgmsa786@gmail.com (G.M.S.A.); 2Department of Mechanical Engineering, G. L. Bajaj Institute of Technology and Management, Greater Noida 201306, India; 3Department of Mechanical Engineering, College of Engineering, King-Khalid-University, P.O. Box 394, Abha 61421, Asir, Saudi Arabia; melkotb@kku.edu.sa (M.A.E.); mmhussien@kku.edu.sa (M.A.H.A.); 4Mechanical Engineering Department, College of Engineering, Kafrelsheikh University, Kafr El-Sheikh 33516, Egypt; 5Center of Excellence (COE) for Advanced Manufacturing Engineering, Program of Mechanical Design and Manufacturing Engineering, Adama Science and Technology University, Adama 1888, Ethiopia; 6School of Engineering, RMIT University, Melbourne, VIC 3001, Australia; bristy808.nh@gmail.com; 7Shoubra Faculty of Engineering, Benha University, Cairo 11629, Egypt

**Keywords:** Al_2_O_3_ nanoparticles, nanofluid, melt fraction, paraffin, PCM

## Abstract

The output of the latent heat storage devices (LHSDs), based on some phase change materials (PCMs), depends upon the thermophysical properties of the phase change material used. In this study, a paraffin-based nanofluid, blended with aluminum oxide (Al_2_O_3_) nanoparticles, is used as PCM for performance evaluation. A three-dimensional (3D) numerical model of regenerative type shell-and-tube LHSD is prepared using COMSOL Multiphysics^®^ 4.3a software to estimate the percentage of melt and the average temperature of the analyzed nanofluids. The results of this study are in close agreement with those reported in the literature, thereby ensuring the validation of the numerically predicted results. The effects of adding the nanoparticles on the rate of melting, as well as solidification and rate of stored/liberated energy, are studied. The results revealed that, by adding 10% nanoparticles of Al_2_O_3_, the melting rate of pure-paraffin-based LHSD improved by about 2.25 times. In addition, the rate of solidification was enhanced by 1.8 times. On the other hand, the heat of fusion and specific heat capacities were reduced, which, in turn, reduced the latent and sensible heat-storing capabilities. From the outcomes of the present research, it can be inferred that combining LHSD with a solar water heater may be used in technologies such as biogas generation.

## 1. Introduction

Energy, a prime entity in any country’s growth, affects the economic development of any country. Conventional energy reserves such as coal, petrol, and natural gas may create environmental hazards such as air pollution, greenhouse emissions, and acid rain. In addition, the combustion of conventional fuels may lead to the emission of particulate matter and chemicals. Hence, there is a need for switching from conventional to alternative energy resources to save the environment. One of the sustainable and potential candidates pertaining to a clean environment is solar energy. However, the intermittent availability of solar radiations restricts its wider utilization. To overcome the barrier of intermittency of solar radiations, using solar devices embedded with integral latent heat storage (LHS) systems, has recently become popular. 

Li et al. [1] conducted the sensitivity analysis of transient, 2D, axisymmetric model of shell-and-tube latent heat (LH) thermal energy storage (TES) and reported an improvement in thermal energy storage of the heat storage system using cascaded PCM. Qaiser et al. [2] studied the effect of stearic acid, as a phase change material (PCM) by circulating it inside the annulus shell (steel) and tubes (copper), containing water as a heat transfer medium, and reported an enhancement in heat transfer rate by 85%. Kalogirou [3] and Bouhal et al. [4] reported that flat plate collectors are economically viable for the conversion of solar energy into useful form without disregarding the environment. 

Rad et al. [5] evaluated the cost-effectiveness and applications of solar heating systems and found them to be economical and suitable options for heating applications. The TRNSYS tool was used to examine and test the performance characteristics of a solar water heating system [6,7]. Gunjo et al. [8,9] numerically and experimentally investigated the performance parameters of solar heating systems. They examined the thermal efficiency and exit water temperature of a solar water heating system computationally and experimentally. They recommended that LHS systems are better than sensible heat storage systems due to their superior storage capacity, high energy density, and high heat of fusion [10]. 

Considering factors such as storage, economy, and availability, Farid et al. [11] proposed that paraffin wax has advantages for a variety of high as well as low-temperature applications. Jesumathy et al. [12] examined the performance characteristics of LHS based on paraffin wax and reported that an increase in the entry temperature of the heat transfer fluid (HTF) reduced the melting time. Rabha and Muthukumar [13] examined the performance of a solar dryer working with paraffin wax and reported an improvement in its performance, even during off-sunshine hours. Agarwal and Sarviya [14] studied the effectiveness of LHS based on paraffin wax for solar drying applications, using water as the HTF, and found them a suitable option for drying. Salunkhe and Krishna [15] conducted a thorough investigation on PCMs for use in LHS systems and emphasized the benefits of using paraffin wax as a storage medium for solar water heating applications. Kabeel et al. [16] compared the efficacy of desalination-system-based LHS with traditional solar still systems. They found that freshwater delivery using an LHS-based system is better than the conventional system. In another study [17], Kabeel et al. also studied a solar air heater, with finned plate, and reported an increase in the efficiency by using an LHS system based on paraffin wax. Niyas et al. [18] explored the consequences of HTF flow rate and temperature on the storing performance of the LHS devices using paraffin wax. 

The findings of Fan and Khodadadi [19,20,21] proposed the techniques of enhancing the conductivity of PCMs with nanoparticle addition, as they improve the thermal properties of the PCM, identical to the reporting of [22,23]. Paraffin-based nanofluids were also applied over carbon nanotubes containing multilayers. A reduction in latent heat and an increase in thermal conductivity were observed [24]. Dheep and Sreekumar [25] reported that nanofluids based on carbon supersede other materials due to their superior phase change behavior. Thapa et al. [26] worked with an energy storage device of small-scale nature and reported that, by adding metallic inserts and copper foams, wax conductivity may be improved. Owolabi et al. [27] examined the use of iron nanoparticles to increase the characteristics of TES, which is used for solar heating applications. They reported that using nanoparticles improved the efficiency by 10%, thereby saving 28.5% in annual costs. Nanofluids for solar thermal applications were also evaluated by Mahian et al. [28]. Said et al. [29] studied the effect of the size of Al_2_O_3_ nanoparticles on thermal storage performance and observed that a particle size of 13 nm led to 73.7 percent storage efficiency. Xie et al. [30,31] investigated the thermal behavior of Al_2_O_3_ nanoparticles volume fraction and reported that thermal conductivity increased with an increase in the concentration of nanoparticles. Xuan et al. [32,33] conducted a theoretical analysis of the thermal behavior of nanofluids containing suspended copper nanoparticles and found it to be an effective way of heat transfer enhancement. Khanafer et al. [34] plotted effective conductivity maps through a numerical examination of heat flow characteristics of nanofluids. Murshed et al. [35] investigated the thermal conductivities of nanofluids seeded with TiO_2_ and found that the thermal conductivity was enhanced by increasing the concentration of nanofluids. 

In this paper, to increase the biogas output from the plant, a 5 MJ capacity LHS solar water heating system was devised and manufactured that acts as a heat source, helping to grow micro-organisms (thermophilic microbes). A three-dimensional (3D) computational domain for LHS with paraffin wax was developed. Since the paraffin wax has a low thermal conductivity, Al_2_O_3_ nanoparticles were dispersed inside the original substance. The effects of heat transfer and cylinder orientation (configuration), along with tube arrangement on the thermal performance, were also studied. 

## 2. Model, Geometry, and Validation

Figure 1 depicts the cross-section of a 3D regenerative, shell-and-tube (SNT) latent heat storage (LHS) model containing water as the heat transfer fluid (HTF) and paraffin wax as phase change material (PCM) seeded with Al_2_O_3_ nanoparticles. The shell diameter and length were 300 mm and 1000 mm, respectively. Radial thickness and the internal diameter of the tubes were fixed as 12 and 3 mm, respectively. The external surface of the shell is insulated to minimize heat loss to the environment. Adopting the optimization technique, the total number of tubes in this paper was selected as 17, and as per the reporting of [36,37], the Al_2_O_3_ nanoparticle addition was limited to 10%.

The assumptions made for the simulations that were conducted using COMSOL Multiphysics^®^ 4.3a [38] were as follows: (i) no agglomeration inside the shell; (ii) flow is laminar, incompressible (Newtonian), and without any viscous heat dissipation; (iii) constant initial temperature; (iv) nanofluid is isotropic and homogeneous; (v) no heat loss from shell’s outer wall; (vi) phase change occurs in a fixed temperature range.

The adopted governing equations were identical to those developed by Niyas et al. [16], concerning the conservation equations of energy, mass, and momentum. The gravity force and source terms, based on Darcy’s law, were considered to determine the velocity of the solid part in the mushy zone (dampening). The source term, as reported in [39] regarding Darcy’s law, encompasses the momentum equation to incorporate the velocity in the solid area of the mushy zone. The mushy region’s velocity transformation can be inferred using mushy zone constants that lie within a range of 10^3^ to 10^7^ [18]. In the present study, the constant was fixed as 10^4^ based on the recommendations of [16]. The parameters such as specific heat (Cp) and effective heat capacity were computed as per the recommendations of [40,41,42]. The HTF received no flow at the starting condition (i.e., at t = 0). No-slip condition at the wall surface was considered. At the inlet, a constant velocity of 0.05 m/s was maintained. The initial temperature was fixed as 298 K in the whole domain. Inlet temperature (Ti) was set as 343 K at the inlet boundary, and the temperature at the outflow was set as 343 K at the exit of the tube. This ensures a negligible temperature slope along the flowing path. Symmetry boundary was employed at the symmetry planes. To minimize heat loss to the surrounding environment, an adiabatic condition (Q = 0) was chosen at the shell’s exterior face. Boussinesq’s approximation was considered to determine the buoyancy effects of the phase change material. The mesh contained tetrahedral cells in the domain (Figure 2). The small section (thickness of HTF tube) was extensively honed and had an adequate volume of elements. For the proposed model, the special meshing capabilities of COMSOL Multiphysics, notably boundary layers, were used to accurately describe boundary layer evolution in the near-wall regime. This feature contributes to the accurate and precise prediction of thermal gradients and velocities near walls. 

The governing equation was discretized using the Galerkin method and finite elements. Inflation layer meshing was carried out at the edges that may face severe gradients. PARDISO [43], a non-linear time-dependent solver that solves the nonlinearity linked with the EHC technique, was chosen. The convergences of temperature and velocities were set as 10^−3^. 

The backward differentiation formula was used as the time-step technique, with the lowest time-step value of 0.01. The grid independence test was conducted (Figure 3). The average temperature showed no change beyond 306,464 elements, which was selected as the final grid size. The thermophysical properties of the paraffin wax and Al_2_O_3_ used in this study are presented in Table 1.

The term “nanofluid” is used throughout this study to refer to paraffin wax dispersed with Al_2_O_3_ nanoparticles. The effective thermal conductivities of nanofluids were calculated using the expressions reported by [37]. The nanofluid’s thermophysical characteristics are affected by nanoparticle concentration, PCM, and temperature [44]. In the present study, suspension density was estimated as per the recommendations of [44], and the PCM’s density was estimated using the formulations reported by [45]. The thermal properties were evaluated as per the recommendations of [45,46]. The definition of the parameters such as solidus temperature, liquidus temperature, melt fraction, charge time, discharge time, etc., dealt with in this paper, can be found in Gunjo et al. [8,9]. The energies (sensible and latent) and the amount of energy released were calculated using equations reported by [47].

## 3. Experimental Device

The external shell of the LHS was made from stainless steel (SS) and was insulated using thermofoam insulation to prevent heat loss to the surrounding. The used shell was filled with 23 kg of PCM (paraffin wax). Water was used as HTF, which passed through the tubes at 343 K or 298 K throughout the charge–discharge duration, allowing heat transfer to take place between the paraffin wax and the HTF. Figure 4 displays the experimental configuration of the LHS as well as an expanded view of the system used for storage. The storage, which was linked to a solar collector of the flat-plate type, was powered with a closed-loop forced movement system. The changing nature of solar radiation, which changes the HTF temperature at the inlet of the LHS, was closely monitored. To this end, a heater, powered electrically with a thermostat, was built inside the reservoir to ensure a uniform temperature of supply at the inlet of the storage system. The flow rate, solar intensity, and temperatures were measured using rotameter pyranometers and T-type thermocouples (Agilent-34972 A with 20 channels multiplexer), respectively (Figure 5). The uncertainty was calculated following the procedures explained by [8,9] and is presented in Table 2. The estimated error, based on the error estimation, corresponded to 7% in the present study.

The results of the computational model were validated by conducting experiments on LHS filled with paraffin wax (Figure 6). The numerical results closely matched the experimental data, and their trends were also identical to the trends reported by [18].

## 4. Results and Discussion

Two distinct configurations were tested in the current optimization study: four tubes in the center and twelve tubes on the outside. The configurations are shown in Figure 7a,b. An initial temperature of 343 K was set for the shell, and the HTF was circulated at 298 K. After discharge, it was observed that the total time taken was less in the configuration shown in Figure 7a than that shown in Figure 7b. Hence, the configuration shown in Figure 7a was selected for the present study. Using an SNT heat exchanger based on a single shell in the LHS device, the effect of orienting the cylinder, as well as the melting fraction of LHS filled with nanofluid during the charging stage, was studied (Figure 8a,b).

It can be seen that, during the first 7 min, melt fraction and average temperature changes in both horizontal and vertical orientations are comparable. After acquiring the melting temperature, the nanofluid’s melt fraction and average temperature increased substantially, indicating a fully charged zone in vertical as well as horizontal configurations. The mean temperature differences between horizontal and vertical configurations were noted as 4.0 K and 2.1 K before and after melting, respectively. In addition, the horizontal configuration reached a fully charged temperature (343 K) and uniform melt fraction in 50 min, which was 20 min faster than the time observed in the vertical system. Hence, the horizontal orientation was adapted for numerical simulation in the present study.

The contours of the melting fraction when it underwent charge and discharge processes are shown in Figure 9. It can be observed that, for full charging, the time was relatively less, compared with the discharging time. The total time it took the nanofluid to fully melt and solidify was 40 and 68 min, respectively. The solidification time of Al_2_O_3_ nanoparticles is faster than CuO nanoparticles, as reported by Gunjo et al. [9]. The figure also indicates that the LHS filled with nanofluids completely melted within 40 min, but the solidification process had not yet been completed. During the charging phase, natural convection heat transfer occurs. Conduction, on the other hand, dominates the discharge process. Consequently, charging takes less time than discharging. During charge–discharge cycle, the HTF temperature was fixed as 343 K and 298 K, respectively. Figure 10a,b depict the consequence of the average temperature (volumetric) while undergoing charge and discharge.

The simulation results revealed that, for nanofluids and pure paraffin, the time required to reach a totally charged state was 40 min and 90 min, respectively. As a result, the use of Al_2_O_3_-dispersed nanofluid is advantageous since it allows for the achievement of a fully melted state 2.25 times quicker than pure paraffin. From Figure 10b, it can also be inferred that the PCM based on pure paraffin wax required 120 min to reach a fully discharged state, whereas the nanofluid required only 68 min (1.76 times less). Furthermore, the average temperature variation followed a similar pattern as that previously reported for nanofluids [26] and LHS systems based on pure paraffin [18].

The melt fraction changed slowly with time, during the early stages (Figure 11). As the nanofluid is initially solid while charging, the primary heat transmission occurs due to molecular diffusion. However, as time passes and phase shift occurs, natural convection comes into play, and eventually, it speeds up the melting process. As demonstrated in Figure 11, the slope of the variability in the melt fraction during the initial phase became significantly sharper for the nanofluid. The improvement in conductivity caused by adding highly conductive nanoparticles is one such mechanism. It was also observed that the PCM, based on pure paraffin wax, required 90 min to completely melt. On the other hand, the nanofluid achieved the same in 40 min, thereby creating a time difference of 50 min between these two processes. The time required to attain a fully discharged condition in the nanofluid was 1.76 times faster than that needed in pure paraffin wax (Figure 11b). As the PCM is liquid at its initial phase, the buoyancy effect prevails, and the natural convection serves as the major cause of heat transfer. As the PCM solidifies over time, conduction heat transmission occurs. The charging period was substantially longer than the discharging period, for both nanofluid and pure paraffin. The sensible energy stored or released throughout the charging or discharging process is indicated in Figure 12. The maximum sensible heat stored for pure paraffin and nanofluid was predicted as 1.37 MJ and 1.8 MJ after the passage of 40 and 90 min, respectively. The results revealed that the net sensible heat stored in the nanofluid was smaller than that of paraffin. The presence of Al_2_O_3_ nanoparticles reduced the specific heat of the PCM. In addition, the stored sensible energy varied linearly with the specific heat. Hence, due to the decrease in sensible heat from 2000 to 1582 J/kg K, the stored sensible heat was reduced. Likewise, the highest sensible heat that was released during discharge was predicted as 1.2 MJ and 1.4 MJ (Figure 12b). This indicates that the nanofluid storage lowered the sensible heat release rate by about 1.45 times, compared with pure paraffin. The maximum latent heat that was stored in the case of pure paraffin and nanofluid after attaining a completely charged condition was 2.2 MJ and 3.44 MJ at 40 min and 90 min, respectively (Figure 13a). Despite the fact that the inclusion of nanoparticles aided in the faster attainment of the charged condition, the total latent heat stored in the case of Al_2_O_3_-dispersed nanofluid was significantly reduced. This is attributed to the fact that there is a linear relationship between latent heat of fusion and net latent heat storage (LHS). Furthermore, the addition of an Al_2_O_3_-based nanoparticle reduced the heat of fusion from 168 to 111 KJ/kg, resulting in a considerable difference in the latent heat storage capability. The melt fractions during the charge and discharge processes are plotted in Figure 13. It can be inferred from Figure 13b that the maximum latent heat released after reaching a completely discharged condition was 3.2 MJ (paraffin) and 2.2 MJ (nanofluid) after 120 min and 68 min, respectively. The trend of variation in total heat with the passage of time during the charge–discharge cycle can be seen in Figure 14. The total energy stored during charging with and without nanoparticles was predicted as 5.24 MJ and 3.57 MJ at 90 and 40 min, respectively (Figure 14a). Further, the total heat released while discharge for paraffin and nanofluid was 3.3 MJ and 4.8 MJ after 68 and 120 min, respectively.

The density did not change with temperature (Figure 15a). However, with an addition of 10% Al_2_O_3_ nanoparticle in the paraffin, the density of the resulting substance increased from 760 to 1031 kg/m^3^. Additionally, adding nanoparticles to paraffin raised the viscosity value from 0.003 Pa-s to 0.013 Pa-s and thermal conductivity from 0.2 W/m K to 0.38 W/m K. As reported earlier, Figure 15d,e show that the specific and latent heat capacities do not dependent upon temperature. Based on the heat capacity data for nanofluids containing 10% Al_2_O_3_, the latent and specific heat capacities were reduced from 168 to 111 KJ/kg and 2000 to 1575 J/kg K, respectively. As a result, the storage ability of the nanofluid-based LHS was reduced, compared with the storage ability of its pure form.

## 5. Conclusions

In this paper, a 3D computational analysis of a shell-and-tube (SNT) LHS system was evaluated. The predicted results, when validated with measured results, showed fairly good agreement with each other. The effects of the addition of nanoparticles of Al_2_O_3_ on the melting rate and storage capabilities were investigated. The results with Al_2_O_3_ addition were compared with the results of the LHS system based on pure paraffin. The following conclusions were drawn: The time required to attain a fully charged condition using nanofluid was reduced by 2.25 times, compared with that needed in the base substance (paraffin). In addition, the time required to attain a fully discharged condition was approximately 1.8 times more rapid. However, compared with the specific heat data of pure paraffin, the specific heat capacity was reduced due to Al_2_O_3_ nanoparticle addition.A reduction in net sensible heat of about 0.43 MJ and 0.21 MJ was observed with Al_2_O_3_ addition during the storage and release processes that occurred during charge and discharge.The latent heat stored or released in the nanofluid was reduced by 1.24 MJ to 1.6 MJ, compared with the primary LHS filled with pure paraffin. This is due to the reduction in the heat of fusion during the process of charge or discharge.By adding 10% Al_2_O_3_ nanoparticles with paraffin, reductions in specific and latent heat capacities of the substance were observed, compared with those of the base fluid. By contrast, increases in the density, viscosity, and thermal conductivity of the substance were detected.

## Figures and Tables

**Figure 1 materials-15-04427-f001:**
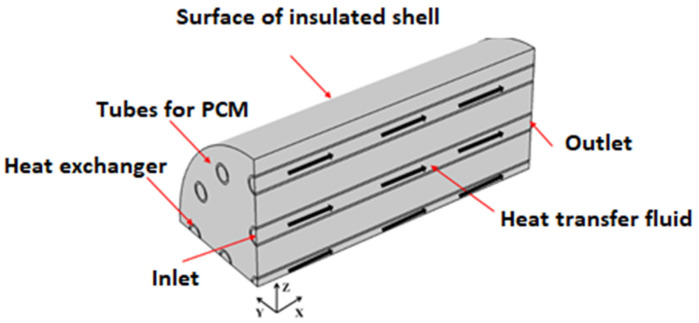
An LHS’s 3D flow domain.

**Figure 2 materials-15-04427-f002:**
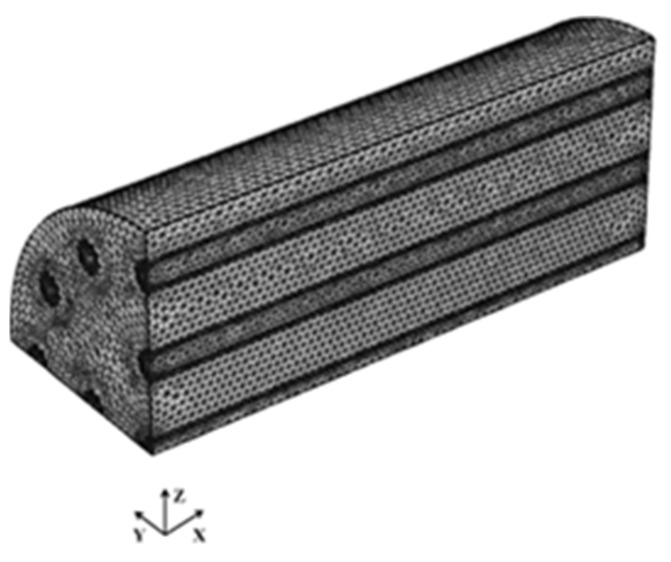
Mesh generated for the used LHS model.

**Figure 3 materials-15-04427-f003:**
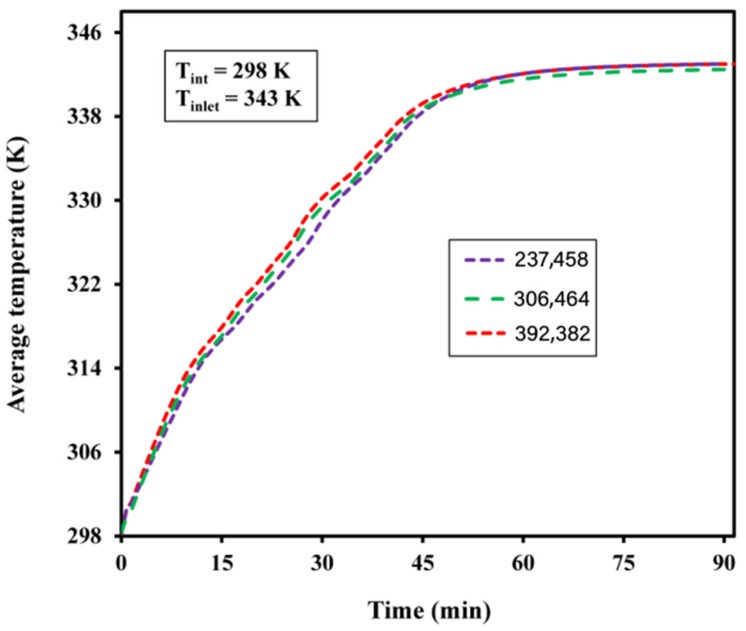
Grid independence check.

**Figure 4 materials-15-04427-f004:**
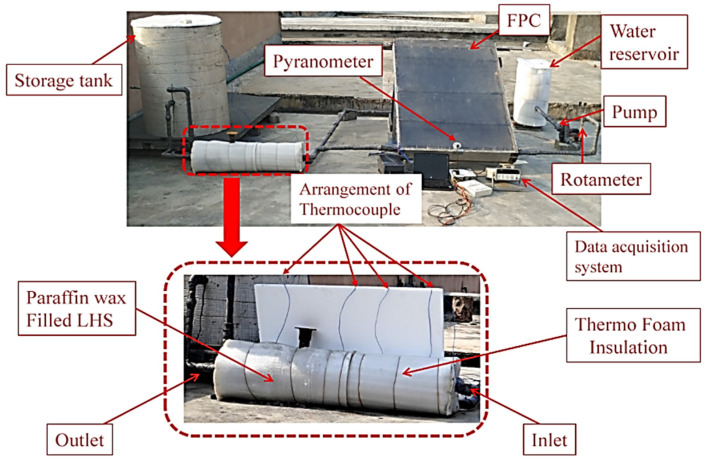
Experimental device with LHS system.

**Figure 5 materials-15-04427-f005:**
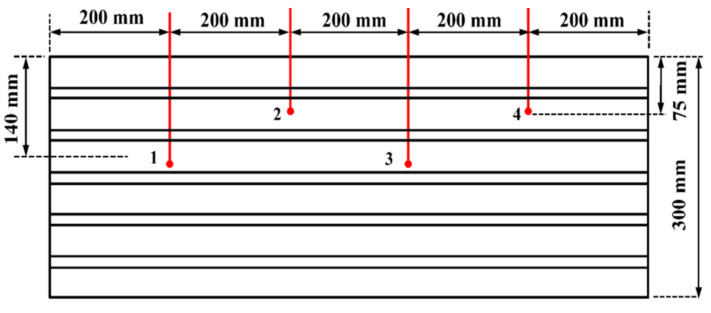
Thermocouple locations (1–4).

**Figure 6 materials-15-04427-f006:**
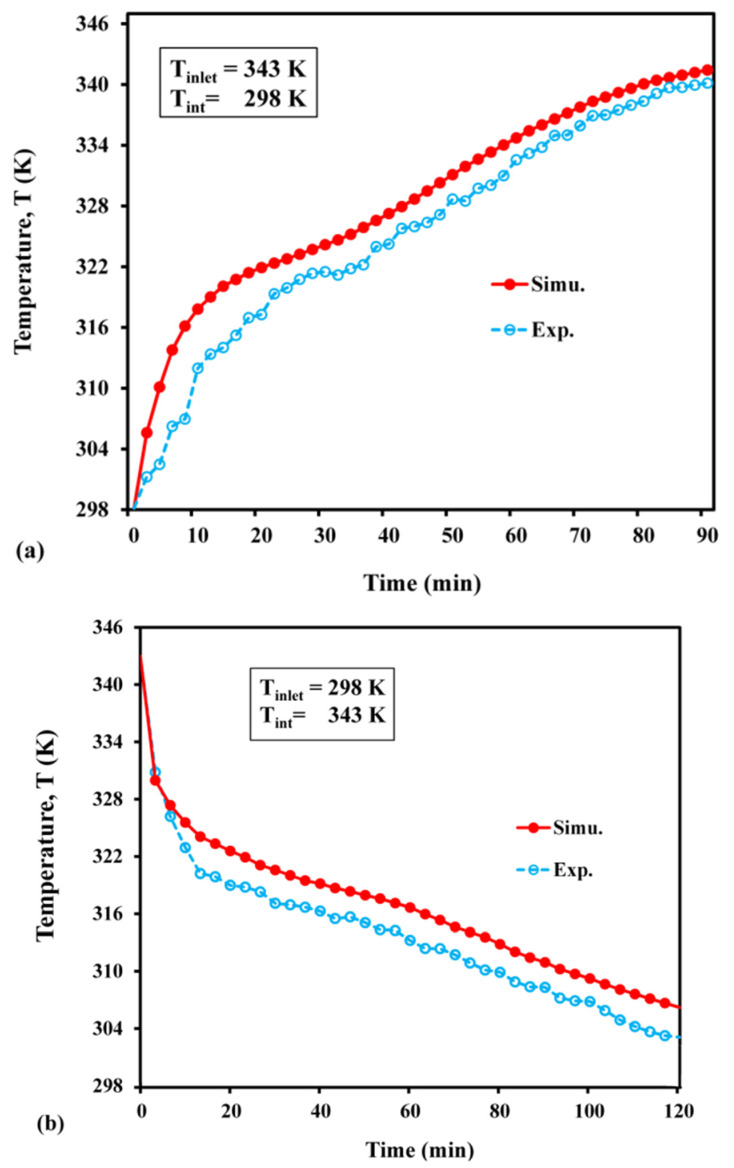
Average temperature distribution: (**a**) during charge; (**b**) during discharge.

**Figure 7 materials-15-04427-f007:**
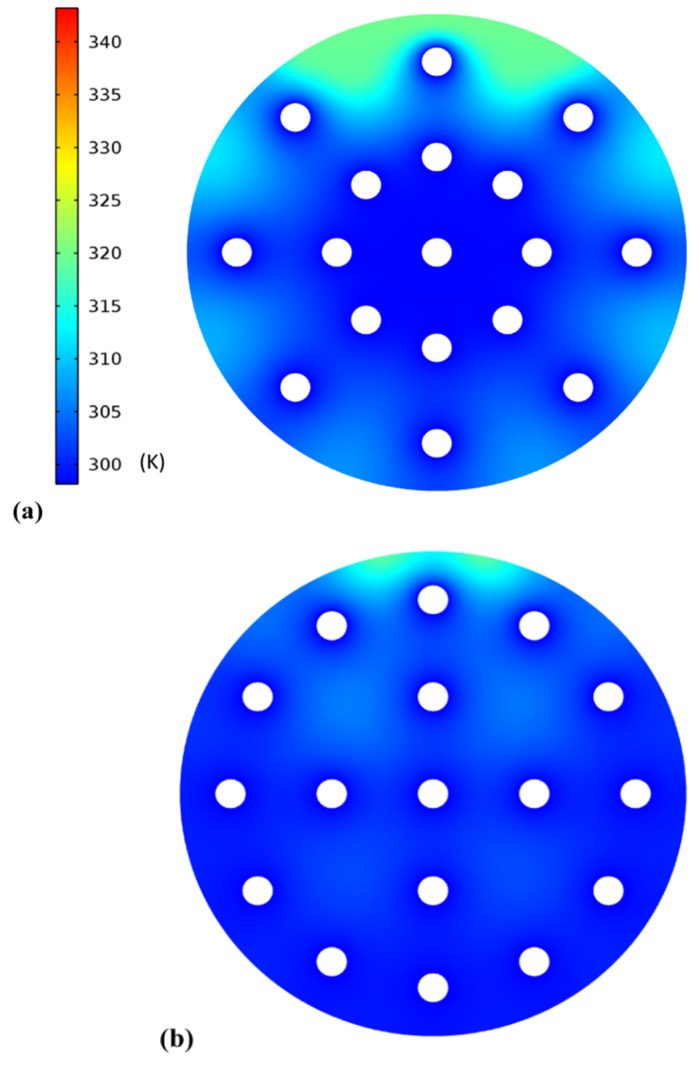
Average temperature varied with tube layout: (**a**) four tubes on inside and twelve tubes on outer side; (**b**) eight tubes on inner side and eight tubes on outer side.

**Figure 8 materials-15-04427-f008:**
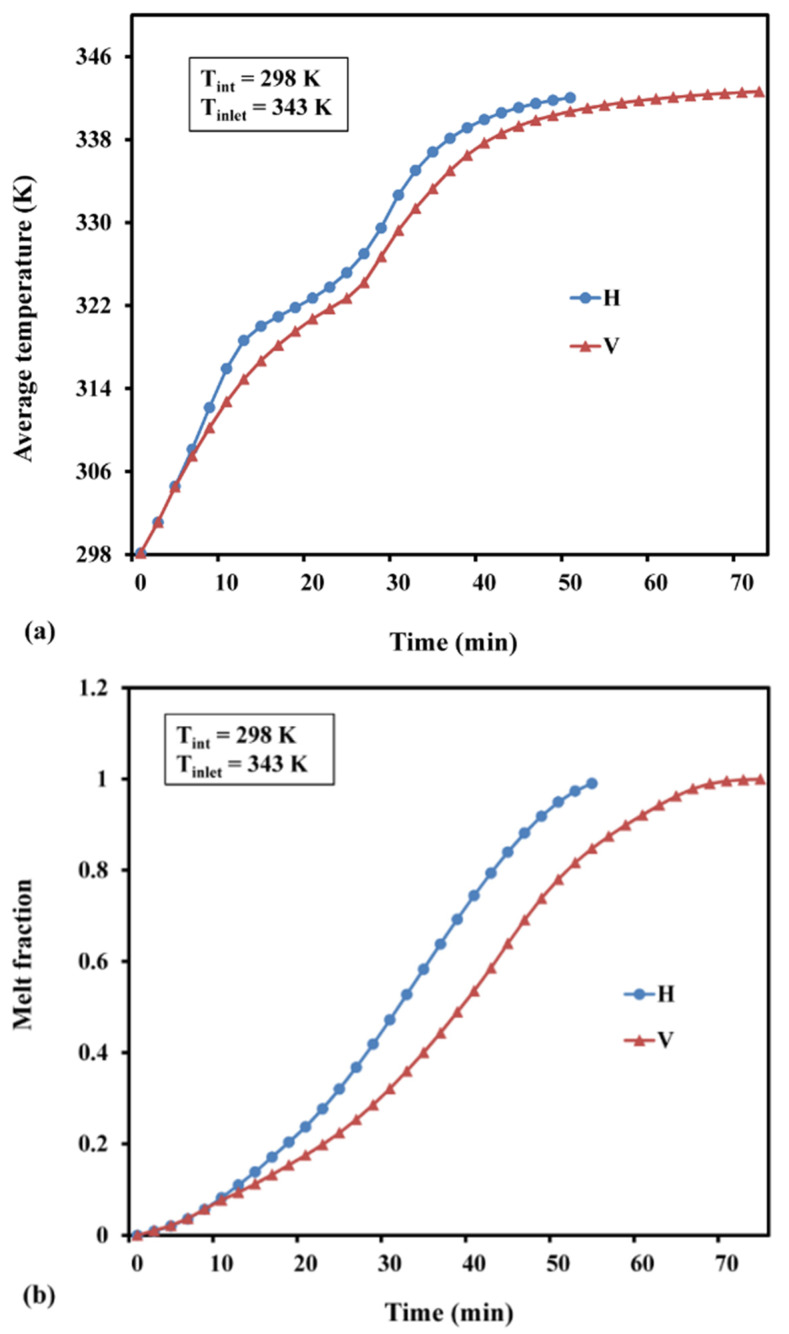
Time dependence of (**a**) average temperature and (**b**) melt fraction (H—horizontal and V—vertical).

**Figure 9 materials-15-04427-f009:**
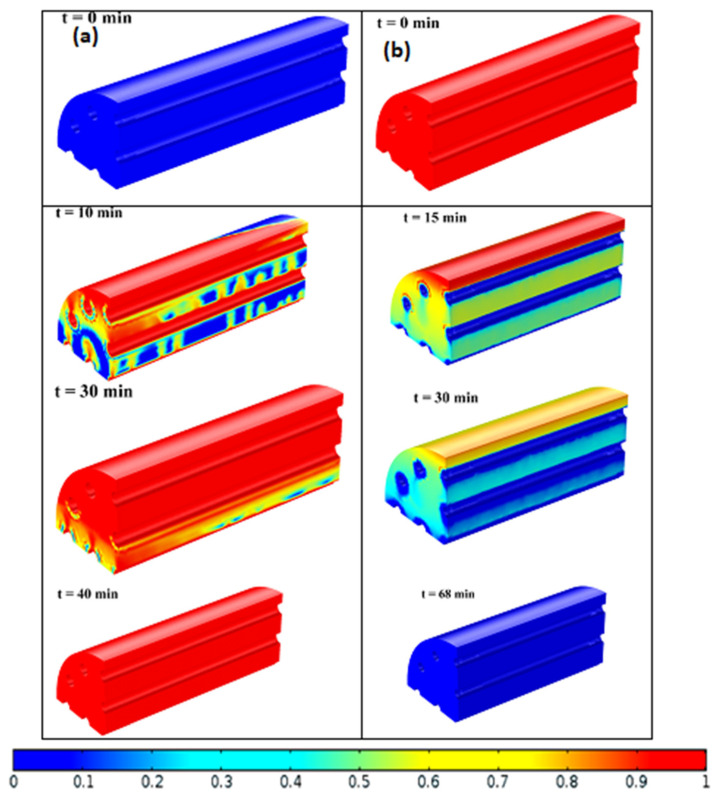
Contours of melt fraction for (**a**) charge and (**b**) discharge processes.

**Figure 10 materials-15-04427-f010:**
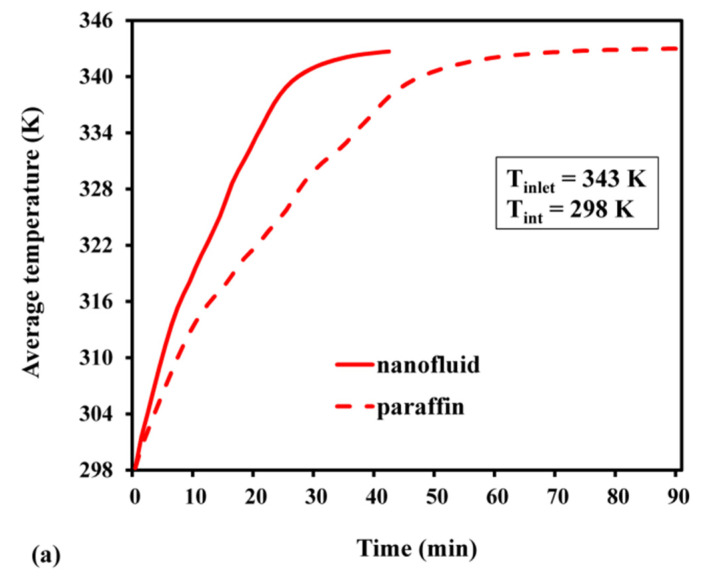
Average temperature: (**a**) during charge; (**b**) during discharge.

**Figure 11 materials-15-04427-f011:**
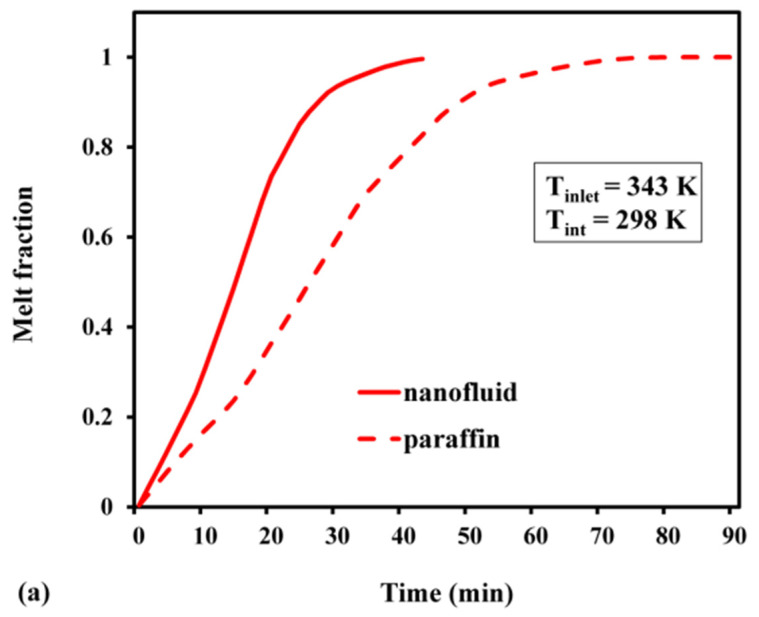
Melt fraction (average): (**a**) during charge; (**b**) during discharge.

**Figure 12 materials-15-04427-f012:**
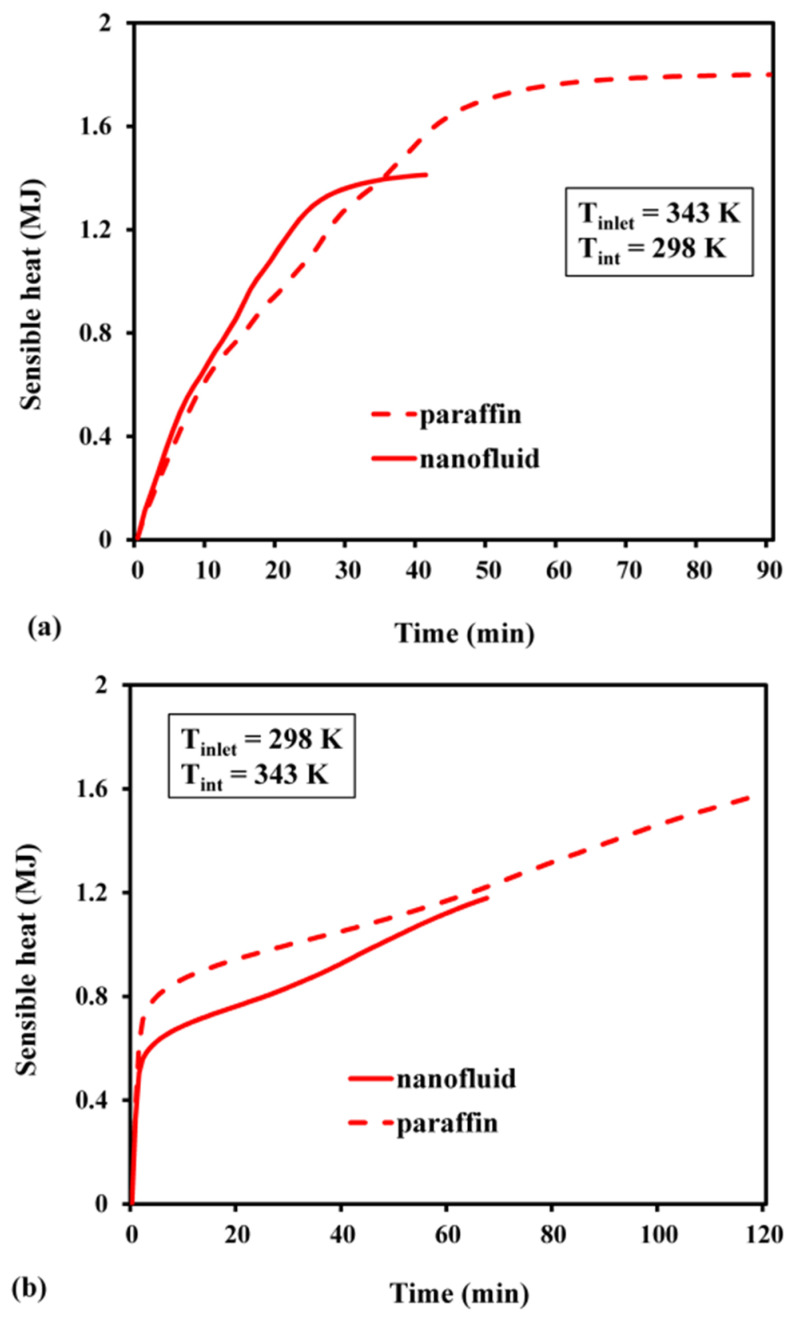
Sensible heat: (**a**) during charge; (**b**) during discharge.

**Figure 13 materials-15-04427-f013:**
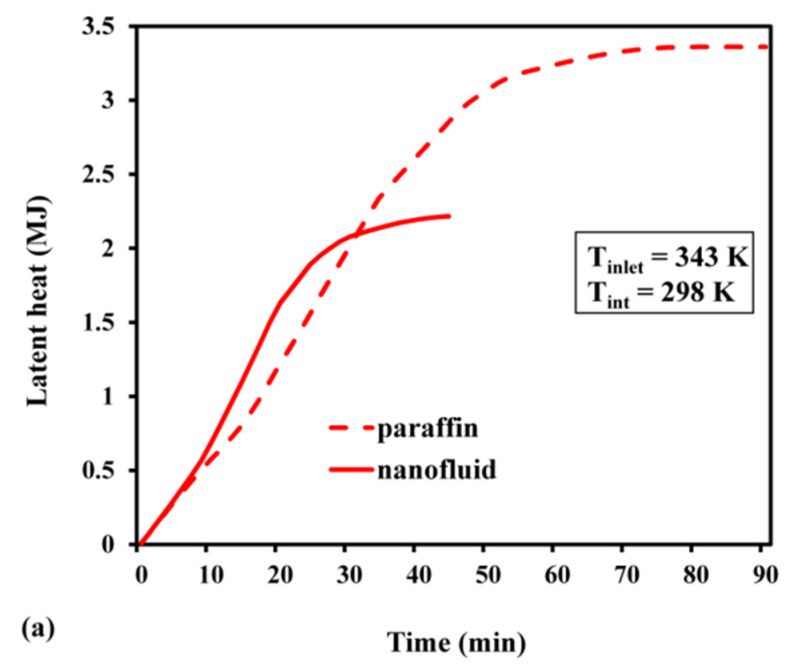
Latent heat: (**a**) during charge; (**b**) during discharge.

**Figure 14 materials-15-04427-f014:**
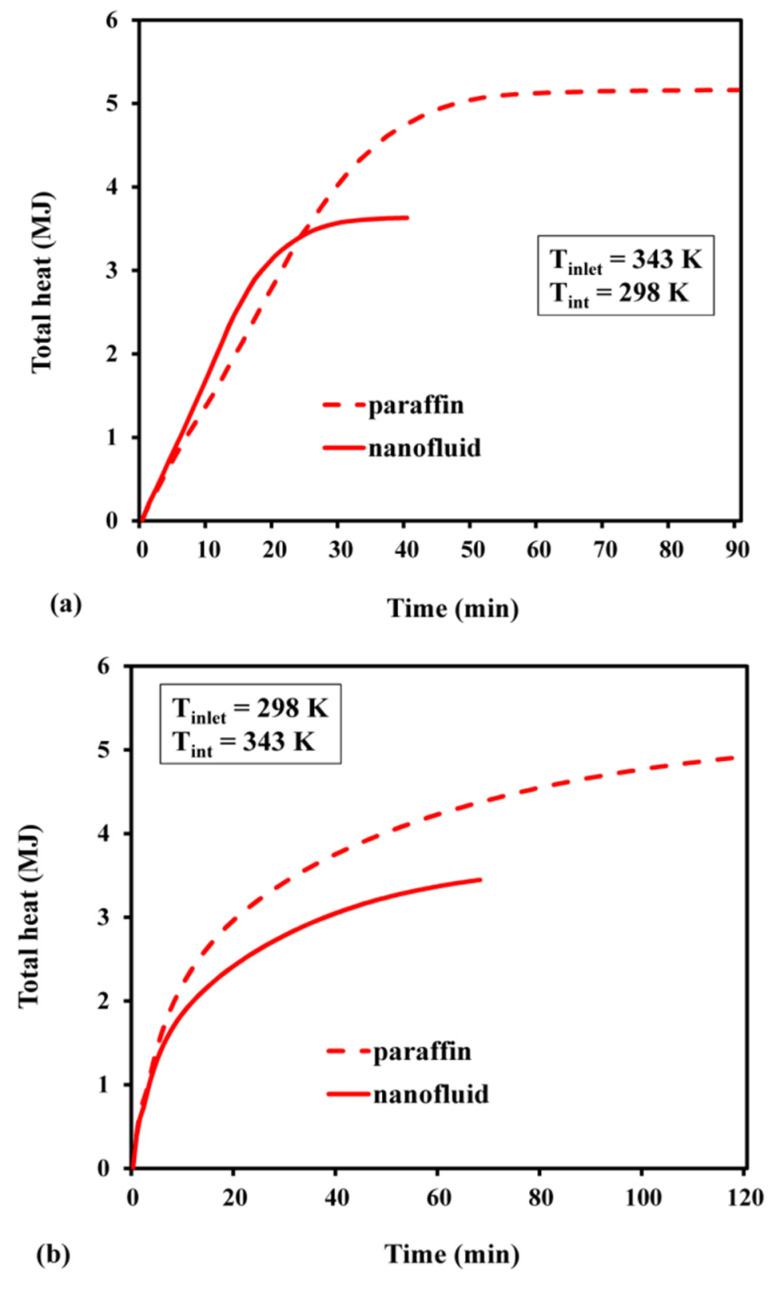
Total heat: (**a**) during charge; (**b**) during discharge.

**Figure 15 materials-15-04427-f015:**
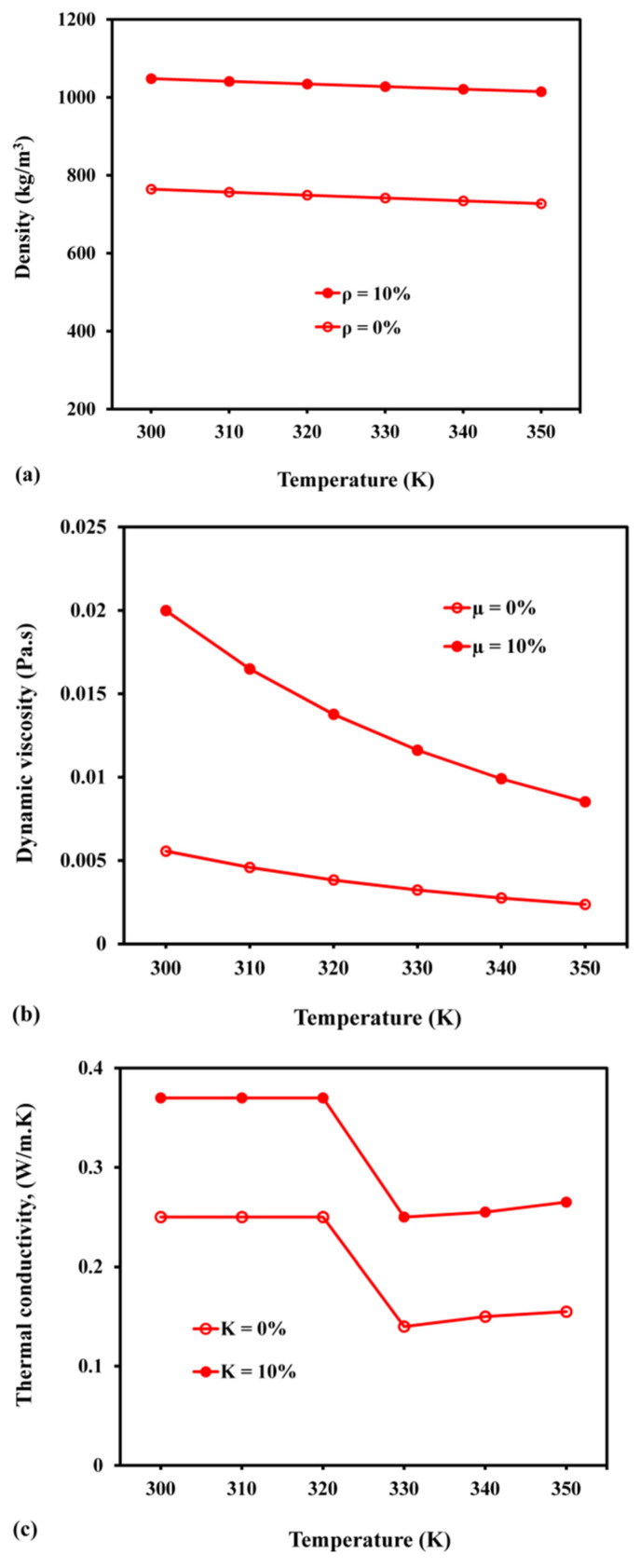
Variations in PCM parameters with temperature: (**a**) density; (**b**) dynamic viscosity; (**c**) thermal conductivity; (**d**) specific heat; (**e**) latent heat.

**Table 1 materials-15-04427-t001:** Thermophysical properties of Al_2_O_3_ and paraffin wax [18,44].

Parameter	Value	Unit
C_Pnp_ (Al_2_O_3_)	765	J/kg K
C_Ppcm_	2000	J/kg K
d_np_ (Al_2_O_3_)	59 × 10^−9^	m
K_np_ (Al_2_O_3_)	36	W/m K
K_pcm_	0.25	W/m K
ρ_np_ (Al_2_O_3_)	3600	kg/m^3^
ρ_pcm_	780	kg/m^3^
L_pcm_	168,000	J/kg
T_ref_	273.15	Kelvin (K)
T_inlet_	343.15	Kelvin (K)
T_intitial_	298.15	Kelvin (K)
T_m_	321.15	Kelvin (K)
dT	3	Kelvin(K)
T_s_	T_m_ − dT	Kelvin (K)
T_l_	T_m_ + dT	Kelvin (K)
C_1_ (Al_2_O_3_)	0.9830	-
C_2_ (Al_2_O_3_)	12.959	-

**Table 2 materials-15-04427-t002:** Uncertainty results regarding independent and dependent parameters.

Parameter	Uncertainty
Tube dia. (HTF)	±0.006 mm
Temperature	±0.2 °C
Solar radiation	±5 W/m^2^

## Data Availability

All the required data are available within the article.

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
