# Peer review of "Improvement in Thermal Storage Effectiveness of Paraffin with Addition of Aluminum Oxide Nanoparticles"

_materials, 2022, doi:10.3390/ma15134427_

Round 1
Reviewer 1 Report
Review report
Manuscript title: Improvement of Thermal Storage Effectiveness with Addition of Aluminum Oxide Nanoparticles
Manuscript ID: materials-1761289
Review comments: The work is good and interesting. The work demonstrates the improvement of thermal storage effectiveness and performance evaluation with addition of Aluminum Oxide Nanoparticles. A three-dimensional (3D) numerical model of regenerative type shell and tube LHSD is prepared COMSOL Multiphysics.It is valuable to readers of this journal. I think it will be published in the current journal after modification the issues raised.
1. In abstract “The results showed that the by adding 10% nanoparticles of Al2O3, the melting rate of pure-paraffin based LHSD improves by about 2.25 times.” Is wrong. It should be “The results have showed that by adding 10% nanoparticles of Al2O3, the melting rate of pure-paraffin based LHSD improves by about 2.25 times.”
2. “It was concluded that by combining LHSD with solar water heater may be used in technologies like biogas generation.” Should be “It is concluded that by combining LHSD with solar water heater may be used in technologies like biogas generation.”
3. In the Introduction part needs to be more explanation and comparison to others research work. Need more references added.
4. The English needs to be rechecked and make it more corrections.

Author Response
TITLE-Improvement of Thermal Storage Effectiveness with Addition of Aluminum Oxide Nanoparticles
Manuscript ID: Materials-1761289
Reviewer-1
Query-1: In abstract “The results showed that the by adding 10% nanoparticles of Al2O3, the melting rate of pure-paraffin based LHSD improves by about 2.25 times.” Is wrong. It should be “The results have showed that by adding 10% nanoparticles of Al2O3, the melting rate of pure-paraffin based LHSD improves by about 2.25 times.”
Reply: Corrections are made as per suggestions of the reviewer. The changes are red highlighted in the revised version of the manuscript.
Query-2: “It was concluded that by combining LHSD with solar water heater may be used in technologies like biogas generation.” Should be “It is concluded that by combining LHSD with solar water heater may be used in technologies like biogas generation.”
Reply: Corrections are made as per suggestions of the reviewer. The changes are red highlighted in the revised version of the manuscript.
Query-3: In the Introduction part needs to be more explanation and comparison to others research work. Need more references added.
Reply: Introduction part is further elaborated. Latest research has been added.
Query-4: The English needs to be rechecked and make it more corrections.
Reply: As per the instructions of the reviewer, the English of the paper is rechecked and revised wherever necessary.

Reviewer 2 Report
This paper studies the improvement of thermal storage effectiveness with the addition of aluminium oxide nanoparticles. Although thermal energy storages are the topic of interest given the global energy scenario, however, this study is outdated and does not contribute to the body of knowledge. There are literally hundreds of papers that use Al2O3 nanoparticles for the thermal performance enhancement of LTESUs. Interestingly the paper discusses the numerical results in detail but the section of numerical methodology is not even mentioned. The LTESU design is not new and the results are trivial i.e. increasing the percentage of nanoparticles increases the melting rate. However, the authors while selecting the optimum percentage of nanoparticles did not take into account the decrease in the energy storage capacity of PCM. The amount of energy storage in PCM plays a critical role while selecting the optimum percentage of NP. All in all this paper does not meet the criteria from the point of view of novelty and significance of the results. Therefore, this paper is not suitable for publication.
Author Response
TITLE-Improvement of Thermal Storage Effectiveness with Addition of Aluminum Oxide Nanoparticles
Manuscript ID: Materials-1761289
Reviewer-2
Query-1: The title needs to be more specific: improvement of thermal storage effectiveness of “what material/device” with addition of Al2O3 nanoparticles.
Reply: The title is revised as per the suggestions of the reviewer.

Reviewer 3 Report
In this manuscript titled “Improvement of thermal storage with addition of aluminum oxide nanoparticles”, the authors performed comprehensive study on the effect of Al2O3 nanoparticles on the efficiency of latent heat storage devices. The overall quality of the manuscript is at satisfactory level while I only have one comment on the title of this manuscript. The title needs to be more specific: improvement of thermal storage effectiveness of “what material/device” with addition of Al2O3 nanoparticles. I would therefore recommend this manuscript to be published after minor revision.
Author Response
TITLE-Improvement of Thermal Storage Effectiveness with Addition of Aluminum Oxide Nanoparticles
Manuscript ID: Materials-1761289
Reviewer-3
Query-1: In order to visualize the content of the image, please modify the layout of the image and place the sub-cells in the image, for example, above the image.
Reply: The layout of the images is modified, as suggested by the reviewer. However, the location of marking a, b, c is not disturbed as it may affect the quality of the image if edited.
Query-2: Some sentences in the text are not normatively on throughout the paper.
Reply: The sentences and the English of the paper is thoroughly checked and necessary corrections are made wherever required.
Query-3: There is incorrect writing format in the article, such as Al2O3 in Fig.12b.
Reply: Suitable corrections are made as suggested
Query-4: Please double-check the format of references, such as references 29 and 31.
Reply: All references are thoroughly checked and corrections are made wherever required.
Query-5: Please add the synthesis steps and material characterization of Al2O3 nanoparticles.
Reply: Synthesis steps and material characterization of Al2O3 nanoparticles are added in the revised version of the manuscript.

Reviewer 4 Report
The authors report the effectiveness of enhancing thermal storage capacity by adding alumina nanoparticles and investigate the effect of adding nanoparticles on melting rate, solidification rate, and energy storage/release rate. This study might give some inspiration to some research fields, such as preparation of materials and energy storage. I suggest it to be published after a major revision by addressing the following issues.
1. In order to visualize the content of the image, please modify the layout of the image and place the sub-cells in the image, for example, above the image.
2. Some sentences in the text are not normatively on throughout the paper.
3. There is incorrect writing format in the article, such as Al2O3 in Fig.12b.
4. Please double-check the format of references, such as references 29 and 31.
5. Please add the synthesis steps and material characterization of Al2O3 nanoparticles.
Author Response
TITLE-Improvement of Thermal Storage Effectiveness with Addition of Aluminum Oxide Nanoparticles
Manuscript ID: Materials-1761289
Reviewer-4
Query-1: Authors need to improve language of the paper. In addition, authors must highlight the scope of the study.
Reply: The sentences and the English of the paper is thoroughly checked and necessary corrections are made wherever required. The scope of the study is also covered.
Query-2: Abstract must provide precise information of the key findings of the present work. I suggest revising the abstract to highlight the novelty of the work.
Reply: As suggested, the abstract is revised to highlight the novelty of the work.
Query-3: Introduction should be improved with the latest published research papers in the same domain.
Reply: Introduction part is further elaborated. Latest research has been added.
Query-4: It is also suggested to discuss Fig. 15 in more detail so that valid conclusions can be drawn from the present work.
Reply: Fig. 15 is discussed in detail in the revised version of the manuscript.
Query-5: The conclusion section also needs to be improved based on the revisions made in the results and discussions section.
Reply: The conclusion section is revised to present the novelty of the work.

Reviewer 5 Report
The paper titled “Improvement of Thermal Storage Effectiveness with Addition of Aluminum
Oxide Nanoparticles” presents relevant information about the potential of Al2O3 phase change material for latent heat storage systems. However, the following points should be addressed before its acceptance for the publication.
1. Authors need to improve language of the paper. In addition, authors must highlight the scope of the study.
2. Abstract must provide precise information of the key findings of the present work. I suggest revising the abstract to highlight the novelty of the work.
3. Introduction should be improved with the latest published research papers in the same domain.
4. It is also suggested to discuss Fig. 15 in more detail so that valid conclusions can be drawn from the present work
5. The conclusion section also needs to be improved based on the revisions made in the results and discussions section.
Overall, the quality of the paper is good, and I recommend for its publication after the corrections as suggested above.
Author Response
TITLE-Improvement of Thermal Storage Effectiveness with Addition of Aluminum Oxide Nanoparticles
Manuscript ID: Materials-1761289
Reviewer-5
Query-1: First of all, please revise your title! It is written in general but your work is very specific!
Reply: The title is revised as per the suggestions of the reviewer.
Query-2: Your Abstract must follow a good structure to represent all your work in a glimpse. However, I cannot see a good presentation of your work!
Reply: As suggested, the abstract is revised to highlight the novelty of the work.
Query-3: The strange thing is you have written all your introduction in one paragraph and never separate to several! Why? Is there any intention! Moreover, it is very hard to follow!
Reply: Thanks for this valuable input. The introduction section is separated in paragraphs. Initially, the intention was to minimize the number of pages. However, now its presentable.
Query-4: I couldn’t see your novelty and why you did such research!
Reply: Novelty is added in abstract, results and conclusion section.
Query-5: Your experimental data must be reported with their errors as well!
Reply: The errors are elaborated and added in the revised version of the manuscript.
Query-6: What are H and V in figure 8?
Reply: H stands for horizontal and V for vertical. The full form is added in the figure to eliminate the confusion.
Query-7: Please report your conclusions in bullet format!!!
Reply: The conclusions are revised as per the suggestions of the reviewer. Its now in bullet form.
Query-8: Please put the Cartesian directions for figure 2!
Reply: As suggested, the Cartesian directions are added in figure 2.
Query-9: Please magnify a part of your geometry to show your meshes in more details! Did you apply inflation layers for your simulations?
Reply: Inflation layers were considered during simulations. The grid independency was also checked at the initial level.
Query-10: Please mention the COMSOL version add a proper reference in your reference list to COMSOL!
Reply: COMSOL version is added and cited properly.

Reviewer 6 Report
Dear Authors,
Many thanks for your work. It was interesting and I feel so happy to see such researches. In my opinion it could be presented in a better way. Please follow my below comments on your work:
1- First of all please revise your title! It is written in general but your work is very specific!
2- Your Abstract must follow a good structure to represent all your work in a glimpse. However, I can not see a good presentation of your work!
3- The strange thing is you have written all your introduction in one paragraph and never separate to several! Why? Is there any intention! Moreover, it is very hard to follow!
4- I couldnt see your novelty and why you did such research!
5- Your experimental data must be reported with their errors as well!
6- What are H and V in figure 8?
7- Please report your conclusions in bullet format!!!
8- Please put the cartesian directions for figure 2!
9- Please magnify a part of your geometry to show your meshes in more details! Did you apply inflation layers for your simulations?
10- Please mention the COMSOL version add a proper reference in your reference list to COMSOL!
Author Response
TITLE-Improvement of Thermal Storage Effectiveness with Addition of Aluminum Oxide Nanoparticles
Manuscript ID: Materials-1761289
Reviewer-6
Query-1: First of all, please revise your title! It is written in general but your work is very specific!
Reply: The title is revised as per the suggestions of the reviewer.
Query-2: Your Abstract must follow a good structure to represent all your work in a glimpse. However, I cannot see a good presentation of your work!
Reply: As suggested, the abstract is revised to highlight the novelty of the work.
Query-3: The strange thing is you have written all your introduction in one paragraph and never separate to several! Why? Is there any intention! Moreover, it is very hard to follow!
Reply: Thanks for this valuable input. The introduction section is separated in paragraphs. Initially, the intention was to minimize the number of pages. However, now its presentable.
Query-4: I couldn’t see your novelty and why you did such research!
Reply: Novelty is added in abstract, results and conclusion section.
Query-5: Your experimental data must be reported with their errors as well!
Reply: The errors are elaborated and added in the revised version of the manuscript.
Query-6: What are H and V in figure 8?
Reply: H stands for horizontal and V for vertical. The full form is added in the figure to eliminate the confusion.
Query-7: Please report your conclusions in bullet format!!!
Reply: The conclusions are revised as per the suggestions of the reviewer. Its now in bullet form.
Query-8: Please put the Cartesian directions for figure 2!
Reply: As suggested, the Cartesian directions are added in figure 2.
Query-9: Please magnify a part of your geometry to show your meshes in more details! Did you apply inflation layers for your simulations?
Reply: Inflation layers were considered during simulations. The grid independency was also checked at the initial level.
Query-10: Please mention the COMSOL version add a proper reference in your reference list to COMSOL!
Reply: COMSOL version is added and cited properly.

Round 2
Reviewer 2 Report
This paper studies the improvement of thermal storage effectiveness with the addition of aluminium oxide nanoparticles. Although thermal energy storages are the topic of interest given the global energy scenario, however, this study is outdated and does not contribute to the body of knowledge. There are literally hundreds of papers that use Al2O3 nanoparticles for the thermal performance enhancement of LTESUs. Interestingly the paper discusses the numerical results in detail but the section of numerical methodology is not even mentioned. The LTESU design is not new and the results are trivial i.e. increasing the percentage of nanoparticles increases the melting rate. However, the authors while selecting the optimum percentage of nanoparticles did not take into account the decrease in the energy storage capacity of PCM. The amount of energy storage in PCM plays a critical role while selecting the optimum percentage of NP. All in all this paper does not meet the criteria from the point of view of novelty and significance of the results. Therefore, this paper is not suitable for publication.
Author Response
Manuscript ID: materials-1761289
Reviewer-2
Query-1: Although thermal energy storages are the topic of interest given the global energy scenario, however, this study is outdated and does not contribute to the body of knowledge. There are literally hundreds of papers that use Al2O3 nanoparticles for the thermal performance enhancement of LTESUs. Interestingly the paper discusses the numerical results in detail but the section of numerical methodology is not even mentioned. The LTESU design is not new and the results are trivial i.e., increasing the percentage of nanoparticles increases the melting rate. However, the authors while selecting the optimum percentage of nanoparticles did not take into account the decrease in the energy storage capacity of PCM. The amount of energy storage in PCM plays a critical role while selecting the optimum percentage of NP. All in all, this paper does not meet the criteria from the point of view of novelty and significance of the results.
Reply: Thanks for the valuable inputs of learned reviewer. It is correct that many papers are available on PCM and performance improvement. These are dealing with similar kind of phase change material but for different applications and with different configurations. This work covers a case study in which the performance of a heat exchanger is evaluated using PCM added with Al2O3 nanoparticles. This will add further insight in the existing literature so that the readers can find more applications of Al2O3 especially in heat exchange applications. The numerical methodology is revised to provide required details of the technique used for simulation. Corrections are made as per suggestions of the reviewer. The changes are red highlighted in the revised version of the manuscript.

Reviewer 4 Report
There are still many typos in the manuscript. In addition, abbreviations should be used when they are present for the first time . Therefore, I suggest a minor revision before the publication.
Author Response
Manuscript ID: materials-1761289
Reviewer-4
Query-1: There are still many typos in the manuscript. In addition, abbreviations should be used when they are present for the first time. Therefore, I suggest a minor revision before the publication.
Reply: Thanks to the reviewer. The authors have rechecked the typos of the manuscript. Abbreviations are defined first thereafter utilized in preceding text.

Reviewer 6 Report
Dear Authors,
Many thanks for providing the revised version. It is much better than your initial version! However, still far from publications!
Please provide high quality figures!
I couldnt find a and b for charging and discharging of figure 9!
Please provide more contours like pressure and temperature for both charging and discharging.
Please provide the full view of your geometry and mesh! not only the domain that you solved and selected as symmetry.
Author Response
Manuscript ID: materials-1761289
Reviewer-6
Query-1: Please provide high quality figures!
Reply: All blur figures and contours are converted to 600 dpi. Now, the quality of the images are good and legible.
Query-2: I couldn’t find a and b for charging and discharging of figure 9!
Reply: Suitable corrections are made in Fig. 9 and it is redesigned with 600 DPI (with .png extension).
Query-3: Please provide more contours like pressure and temperature for both charging and discharging.
Reply: Thanks to learned reviewer for valuable feedback. The contours of the melt fraction are represented in Fig. 9 at their respective temperatures. Plotting the pressure and temperature contour may further add the redundancy and bulkiness to the paper. Already there are 15 figures with many sub figures. As a consequence, the length of the paper will become too large. Fig. 7 may also be referred for average temperature contour. The authors request the reviewer to accept the manuscript with melt fraction figure mentioned with temperature. The pressure contour will not help in any major interpretation, hence, it is not included.
Query-4: Please provide the full view of your geometry and mesh! not only the domain that you solved and selected as symmetry.
Reply: If the symmetry is suitably chosen, there is no need to show complete domain. To reduce the computational time and effort, the authors have presented the optimized geometry and domain. The symmetries are simply the mirror image of the chosen geometry that can be easily visualized. Hence, the authors have only shown the key part of the geometry. On further adding the figures in the current manuscript, the number of figures will be more. We request the learned reviewer to kindly accept the figures presented in their present form.
